# Peripheral Coronary Artery Circulatory Dysfunction in Remote Stage Kawasaki Disease Patients Detected by Adenosine Stress ^13^N-Ammonia Myocardial Perfusion Positron Emission Tomography

**DOI:** 10.3390/jcm11041134

**Published:** 2022-02-21

**Authors:** Kanae Tsuno, Ryuji Fukazawa, Tomonari Kiriyama, Shogo Imai, Makoto Watanabe, Shinichiro Kumita, Yasuhiko Itoh

**Affiliations:** 1Department of Pediatrics, Nippon Medical School, 1-1-3 Sendagi, Bunkyo-ku, Tokyo 113-8603, Japan; kanaetsuno@nms.ac.jp (K.T.); s7103@nms.ac.jp (M.W.); yasuhiko@nms.ac.jp (Y.I.); 2Department of Radiology, Nippon Medical School, 1-1-3 Sendagi, Bunkyo-ku, Tokyo 113-8603, Japan; s7026@nms.ac.jp (T.K.); shogoimai@nms.ac.jp (S.I.); s-kumita@nms.ac.jp (S.K.)

**Keywords:** Kawasaki disease, PET, coronary circulation, coronary aneurysm, coronary artery remodeling

## Abstract

Coronary peripheral circulatory disturbances in the remote stage of Kawasaki disease have been reported. In this study, of the 50 patients in the remote stage of Kawasaki disease who underwent coronary perfusion evaluation using adenosine-loaded ^13^N-ammonia positron emission tomography, 28 patients who did not have stenosis of ≥75% in the left coronary artery underwent an evaluation for myocardial flow reserve (MFR) of the left anterior descending artery (LAD) and left circumflex artery (LCx). Clinical findings were compared between patients with normal (≥2.0) and abnormal (<2.0) MFRs. In the group with an abnormal MFR in the LAD, the responsiveness of the coronary vascular resistance to adenosine stress decreased even in the LCx (3.50 ± 1.23 vs. 2.39 ± 0.25, *p* = 0.0100). In the group with an abnormal MFR in the LCx, the responsiveness of the coronary vascular resistance in the LAD also decreased (3.27 ± 1.39 vs. 2.03 ± 0.25, *p* = 0.0105), and the age of onset of Kawasaki disease tended to be younger in the group with abnormal MFR in the LAD and LCx. We found that the peripheral coronary circulation was extensively impaired in the remote stage of Kawasaki disease, suggesting that an early onset of Kawasaki disease may affect the peripheral coronary circulation in later years.

## 1. Introduction

A problem with coronary artery lesions in the remote stage of Kawasaki disease (KD) is the appearance of stenotic lesions due to ongoing vascular remodeling [1]. In addition to the morphological assessment by coronary angiography, the fractionated flow ratio (FFR) has recently been recommended for the evaluation of stenotic lesions, and it has been reported that percutaneous coronary intervention (PCI) based on FFR is superior to PCI based on morphological assessment in terms of morbidity and mortality [2]. FFR is based on the pressure difference before and after coronary artery stenosis under the administration of coronary dilators, whereas coronary flow reserve (CFR) is a method of assessing the increase in coronary blood flow before and after the administration of coronary dilators. Using cardiac catheterization with a coronary Doppler wire, we found that the normal value of CFR in children is over 2.0, similar to that in adults [3]. We have previously reported how FFR and CFR are useful for detecting myocardial ischemia in children with KD cardiac sequelae [3]. Recently, myocardial blood flow (MBF) per myocardial weight has been measured using positron emission tomography (PET) and by measuring MBF before and after the administration of coronary artery dilators, the myocardial flow reserve (MFR) can be evaluated. MFR has the same physiological significance as CFR, and it has been reported that the risk of cardiovascular death is increased in patients with reduced MFR [4]. As decreased MFR was reported in remote stage KD patients [5,6], distal microcirculation may undergo remodeling and compromise myocardial perfusion in the remote stage of KD. We used a ^13^N-ammonia PET scan to evaluate coronary arteries, calculate the MFR, and assess the coronary circulation in remote stages of KD.

## 2. Materials and Methods

This retrospective study included 50 patients diagnosed with KD with coronary artery complications who underwent an adenosine loading ^13^N-ammonia PET scan between July of 2016 and May of 2021 to search for myocardial ischemia. This study was approved by the Nippon Medical School Ethics Committee and performed in accordance with the ethical standards of the Declaration of Helsinki.

All patients had undergone coronary angiography or coronary computed tomography (CT) within two years of this examination, and those with morphological stenosis of 75% or more in the left anterior descending (LAD) or circumflex (LCx) branches were excluded from this study. MBF was evaluated before and after adenosine loading, and the MFR was calculated. The clinical characteristics of enrolled patients were examined and compared after dividing them into two groups: those with MFR ≥ 2.0 and those with MFR < 2.0. The following information was collected: sex; age in months at the onset of KD; age in months at the time of PET examination; number of months elapsed from the onset of KD to the time of PET examination; height, weight, and body mass index (BMI) at the time of PET examination; maximum diameter of the LAD aneurysm; maximum diameter of the LCx aneurysm; presence of >75% stenosis or occlusion of the right coronary artery (RCA); the presence of a major adverse cardiac event (MACE). Coronary artery calcification was also examined in patients who underwent coronary CT within two years of the PET scan. Coronary artery calcification was evaluated as follows: grade 0, no calcification; grade 1, petechial calcification at the aneurysm site; grade 2, diffuse calcification in the area of the mass but not the total circumference of the aneurysm; and grade 3, calcification in the entire circumference of the aneurysm.

### 2.1. Adenosine Stress ^13^N-Ammonia PET Scan

The protocol for adenosine-stress ^13^N-ammonia PET scan is presented in Figure 1. After CT scan of the chest for attenuation correction, 7.4 MBq/kg of ¹^3^N-ammonia tracer was administered via the right elbow vein within 30 s, and 10 min of data acquisition was started simultaneously with administration. MBF quantification by compartment model analysis was performed using the data collected for a total of 4 min (6 s × 20, 30 s × 2, and 60 s × 1) after administration. The absolute value of the MBF was calculated from the time–activity curves of the counts of tracer flowing into the blood pool in the myocardium and the counts of tracer ingested into the myocardium, using a 1-issue (intravascular-myocardial) two-compartment model. Left ventricular function analysis with synchronous ECG acquisition was also performed during the last 5 min of imaging. Adenosine loading was started 40–50 min (4–5 half-lives) after the resting imaging was performed. Adenosine was administered at 144 μg/kg/min via the left elbow vein over 5 min, while ECG, blood pressure, and oxygen saturation were monitored. Three minutes after the start of adenosine administration, a ^13^N-ammonia tracer was administered using the same procedure as that in the resting state, and imaging was performed using the same protocol as that in the resting state.

The MBF of the LAD and LCx at rest and during adenosine stress, and the MFR, which is the MBF ratio between the stress and resting states, were calculated. Additionally, the diastolic blood pressure of the median artery measured at rest and during adenosine stress was substituted as an approximation of the coronary blood pressure. The vascular resistance of the LAD and LCx was calculated as follows:

LAD vascular resistance = diastolic pressure/LAD MBF; LCx vascular resistance = diastolic pressure/LCx MBF.

The MFR of the LAD and LCx was divided into two groups: a normal MFR group (≥2.0) and an abnormal MFR group (<2.0).

The MBF, coronary vascular resistance, and coronary vascular resistance ratio (coronary vascular resistance at rest/coronary vascular resistance at stress) at rest and during stress were compared.

### 2.2. Statistics

Statistical data were expressed as median (upper and lower quartiles) or mean ± standard deviation. The Wilcoxon test was used to compare the two groups. The chi-square test was used to classify the degree of coronary artery calcification, and Fisher’s exact test was used to compare the presence of RCA stenosis and MACE. *p* < 0.05 was considered statistically significant.

## 3. Results

From July of 2016 to May of 2021, 50 patients with KD with coronary artery complications who underwent an adenosine stress ^13^N-ammonia myocardial perfusion PET scan to search for the presence of myocardial ischemia were included in the study. Of these, 28 patients were included in the analysis excluding those with >75% stenosis of the LAD or LCx. The median age of the patients at the time of examination was 251 months (167.5, 294.8), and the median time since the onset of KD was 226 months (152.3, 258.0). The median BMI was 20.2 (16.2, 24.9). Twenty patients had stenosis or occlusion of the RCA, and three patients had MACE (acute myocardial infarction in all cases).

Coronary artery calcification was assessed in 22 patients who underwent CT within two years. The grades of coronary artery calcification in the LAD group were grade 0, 1, 2, and 3 in six, five, one, and 10 patients, respectively, and those in the LCx group were grade 0, 1, 2, and 3 in 10, four, 0, and eight patients, respectively.

### 3.1. Adenosine Stress ^13^N-Ammonia PET Scan

The MFR of the LAD was 2.54 ± 0.95 and that of the LCx was 2.67 ± 0.70 in the patients. The MFRs of the LAD and LCx were classified into two groups: normal MFR (≥2.0) and low MFR (<2.0). The MFRs of the LAD and LCx were compared between the normal and abnormal MFR groups.

The analysis of MFR of the LAD is shown in Table 1. In the resting state, there was no difference in the blood flow and vascular resistance of the LAD and LCx between the normal and abnormal MFR groups. MBF of the LAD and LCx were both significantly increased under adenosine stress in the normal compared with the abnormal MFR group (LAD: 2.73 ± 1.06 vs. 1.69 ± 0.60 mL/g/min, *p* = 0.0100; LCx: 2.36 ± 0.54 vs. 1.72 ± 0.47 mL/g/min, *p* = 0.0187). Coronary vascular resistance was significantly lower in the normal MFR group (LAD: 18.4 ± 8.0 vs. 27.3 ± 6.0, *p* = 0.0161; LCx: 20.2 ± 7.7 vs. 26.5 ± 7.6, *p* = 0.0500). The coronary vascular resistance ratio at rest and under adenosine stress, in both the LAD and LCx, significantly decreased in the MFR-abnormal group (LAD: 3.37 ± 1.41 vs. 2.06 ± 0.21, *p* = 0.0017; LCx: 3.50 ± 1.23 vs. 2.39 ± 0.25, *p* = 0.0100) (Figure 2). In addition, the MFR of the LCx was significantly decreased in the LAD MFR-abnormal group (1.84 ± 0.22 vs. 2.90 ± 0.60, *p* = 0.0004).

### 3.2. MFR in LCx

There were no significant differences in MBF and coronary vascular resistance in the LAD and LCx between the normal and abnormal MFR groups of the LCx during the resting state (Table 2). No significant differences were detected in MBF and coronary vascular resistance in the LCx under adenosine stress between the two groups. MBF in the LAD was significantly higher in the normal MFR group of LCx under adenosine stress (LAD: 2.66 ± 1.05 vs. 1.61 ± 0.75 mL/g/min, *p* = 0.0489) than in the abnormal MFR group, but there was no significant difference in coronary vascular resistance.

However, the coronary vascular resistance ratio at rest and adenosine stress significantly decreased in the LCx MFR-abnormal group (LCx: 2.39 ± 0.28 vs. 3.41 ± 1.21, *p* = 0.0418, LAD: 2.03 ± 0.25 vs. 3.27 ± 1.39, *p* = 0.0105) (Figure 3). The MFR of the LAD also significantly decreased in the LCx MFR-abnormal group (1.47 ± 0.43 vs. 2.72 ± 0.60, *p* = 0.0025).

The LCx MFR-abnormal group tended to have a younger age of onset of KD (11.5 ± 9.6 vs. 32.0 ± 21.6 months, *p* = 0.0707). However, there were no significant differences between the normal and abnormal MFR groups of the LCx in other parameters such as the number of months since the onset of KD, BMI, maximum diameter of the coronary aneurysm, degree of calcification, presence of RCA stenosis or occlusion, and presence of MACE.

## 4. Discussion

The FFR and CFR (clinically treated as synonymous with MFR) are physiological assessment methods for coronary circulation. It is believed that interventions based on FFR (FFR < 0.8), rather than a morphological assessment alone, can lead to decreased morbidity and mortality [7,8]. Therefore, FFR is now widely used to intervene in the case of stenotic coronary arteries in ischemic heart disease. However, because ischemic heart disease is a condition in which the blood flow to the peripheral myocardium is significantly decreased, FFR, which primarily assesses the degree of stenosis in the epicardial artery, is also considered to be a surrogate marker for assessing blood flow, since it evaluates the pressure difference at the stenosis site [9]. In contrast, CFR is an index that considers the influence of blood flow in the peripheral coronary circulation, in addition to the flow in the epicardial artery [10,11]. Therefore, a dissociation between FFR and MFR, a measure of the coronary circulation (i.e., no stenosis in the epicardial artery), suggests an impairment of the peripheral coronary circulation [11,12]. A higher 10-year incidence of MACE has also been reported when FFR is normal and CFR is abnormal [11].

In this study, we examined patients with reduced MFR excluding those with >75% stenosis in the LAD and LCx, and found that patients with a reduced MFR in the LAD and LCx had a reduced response to coronary dilators. The presence of coronary artery microcirculation disturbance was confirmed even in the remote stage of KD, and the disturbance extended to the total left coronary artery (LCA). Although there are many reports of continued endothelial cell dysfunction in the remote stage of KD [13], there are few reports that have directly demonstrated peripheral circulatory dysfunction. Therefore, we believe that this paper is particularly valuable. We also found that patients with a decreased MFR in the LAD and LCx regions tended to be younger at the time of KD onset, suggesting that coronary arteritis at a younger age may affect the development of coronary microcirculation.

Previous reports of PET with H_2_^15^O in patients with pre-existing KD have shown that the increase in blood flow in response to dipyridamole and adenosine stress was significantly reduced in patients with coronary aneurysms [14] and regressed coronary aneurysms [15]. In addition, Muzik et al. [6] performed ^13^N-ammonia PET in patients with KD without cardiac complications and reported that the increase in MBF in response to adenosine loading was reduced compared with that in healthy subjects. In these reports, the MBF was assessed as a whole heart and not by the coronary artery branches such as in this study. In addition, Muzik et al. reported a MFR of 3.2 ± 0.7 in patients with KD assessed by ^13^N-ammonia PET, which was significantly lower than that of normal controls in their study, but was not considered abnormal. In this study, we showed that there were cases with abnormal MFR and abnormal peripheral coronary circulation, even in the remote stage of KD. Alternatively, this could read, “We also demonstrated that an abnormal MFR of the LAD and an abnormal MFR of the LCx were associated with a decreased vasodilating capacity of the peripheral blood vessels of the LCx and LAD regions, respectively”. This suggests that there is an abnormality in the entire LCA, rather than just a region-specific abnormality in the LAD or LCx. Furthermore, KD tended to occur at a younger age in these cases, and the possibility that the severe coronary arteritis in childhood impaired the development of the peripheral coronary arteries may explain the presence of impaired peripheral coronary circulation throughout the LCA.

## 5. Conclusions

Adenosine stress ^13^N-ammonia myocardial perfusion PET revealed the presence of reduced peripheral coronary circulation in the remote stage of KD, suggesting that this may be related to the onset of KD at a younger age. Future studies are needed to determine the type of treatment that is effective in improving peripheral coronary circulation.

### Study Limitations

The study included only 28 patients. We hope that the number of cases will increase in the future.

Only LCAs could be assessed in this study. There were only three cases with RCA MFR < 2.0. In addition, quantitative analysis of RCA territory often demonstrated apparently erroneous values due to overflow from the tracer in the blood pool of the right ventricle or due to artifacts from respiratory motion. For this reason, we could not include RCA in the analysis.

The Doppler wire (FloWire XT, Cardiometrics Inc., Mountein View, CA, USA) required for CFR evaluation is currently difficult to obtain, and without Doppler wire, CFR cannot be performed in many cases. Thus, it is believed that CFR will be replaced by MFR in the future. Furthermore, FFR, which evaluates stenosis in the epicardial artery, was not examined; however, since patients with stenosis of 75% or more on coronary angiography were excluded, it is unlikely to affect the results of this study.

As shown in this study, it is presumed that damage to the peripheral coronary arteries may extend to the entire myocardium; hence, it will be necessary to accumulate more cases and to improve the accuracy of analysis and examine abnormalities of the MFR in the RCA territory separately in the future.

## Figures and Tables

**Figure 1 jcm-11-01134-f001:**
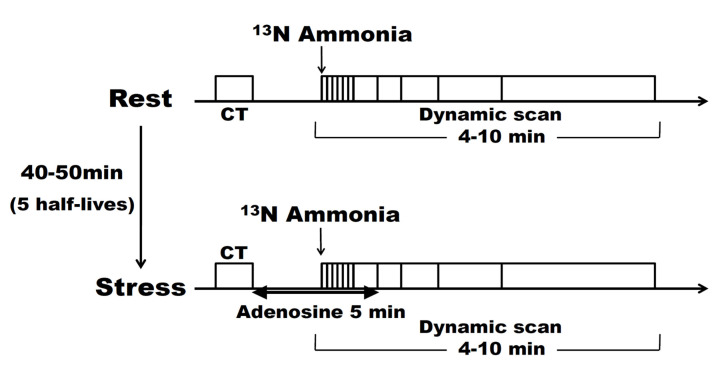
Protocol of the adenosine stress ^13^N-ammonia PET scan. The same protocol was used for both rest and loading, except for the administration of adenosine, with 4–5 half-lives between the rest and loading examinations. CT: computed tomography.

**Figure 2 jcm-11-01134-f002:**
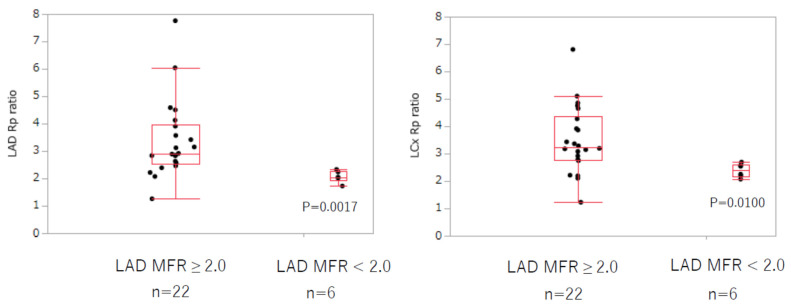
LAD and LCx coronary vascular resistance ratio according to the MFR of the LAD. There was also a tendency for the age at onset of KD to be lower in the LAD MFR-abnormal group (14.5 ± 13.2 vs. 33.0 ± 21.8 months, *p* = 0.0728). However, there were no significant differences between the groups with normal and abnormal MFR in the LAD in other parameters such as the number of months since the onset of KD, BMI, maximum diameter of the coronary aneurysm, degree of calcification, presence of RCA stenosis or occlusion, and presence of MACE. The LAD coronary vascular resistance ratio was significantly decreased in the LAD MFR-abnormal group (MFR < 2.0). The LCx coronary vascular resistance ratio also decreased in the LAD MBF-abnormal group. LAD: left anterior descending artery, LCx: left circumflex artery, Rp: coronary vascular resistance, MFR: myocardial flow reserve.

**Figure 3 jcm-11-01134-f003:**
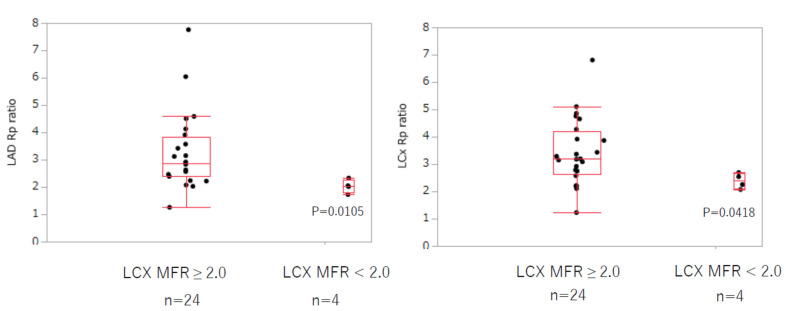
LAD and LCx coronary vascular resistance ratio according to LCx MFR. The LCx coronary vascular resistance ratio was significantly decreased in the LCx MFR-abnormal group (MFR < 2.0). The LAD coronary vascular resistance ratio also decreased in the LCx MFR-abnormal group. LAD: left anterior descending, LCx: left circumflex, Rp: coronary vascular resistance, MFR: myocardial flow reserve.

**Table 1 jcm-11-01134-t001:** Analysis by MFR of the LAD.

	LAD MFR ≥ 2.0	LAD MFR < 2.0	*p* Value
Cases	22	6	
Sex (M/F)	16/6	6/0	
Age of KD onset (months)	33.0 ± 21.8	14.5 ± 13.2	0.0728
Age of PET Exam (months)	252.6 ± 83.2	263.2 ± 153.0	0.8011
Period from KD onset to PET Exam (months)	219.1 ± 90.9	248.8 ± 143.9	0.8666
BMI	19.9 ± 2.8	21.6 ± 2.1	0.1701
At Rest			
systolic BP (mmHg)	97.1 ± 10.3	103.2 ± 19.0	0.5374
diastolic BP (mmHg)	51.4 ± 8.3	56.2 ± 8.6	0.3543
LAD MBF (mL/min/g)	1.00 ± 0.39	1.03 ± 0.21	0.2628
LCx MBF (mL/min/g)	0.83 ± 0.19	0.94 ± 0.08	0.3412
LAD vascular resistance	55.5 ± 16.3	55.6 ± 10.7	0.7369
LCx vascular resistance	64.0 ± 13.5	63.0 ± 17.6	0.9108
Adenosine Stress			
systolic BP (mmHg)	93.8 ± 18.1	98.3 ± 19.8	0.5194
diastolic BP (mmHg)	45.8 ± 15.1	44.2 ± 13.0	0.7579
LAD MBF (mL/min/g)	2.73 ± 1.06	1.69 ± 0.60	0.0100
LCx MBF (mL/min/g)	2.36 ± 0.54	1.72 ± 0.47	0.0187
LAD vascular resistance	18.4 ± 8.0	27.3 ± 6.0	0.0161
LCx vascular resistance	20.2 ± 7.7	26.5 ± 7.6	0.0500
Coronary vascular resistance ratio (rest/adenosine stress)			
LAD	3.37 ± 1.41	2.06 ± 0.21	0.0017
LCx	3.50 ± 1.23	2.39 ± 0.25	0.0100
LCx MFR	2.90 ± 0.60	1.84 ± 0.22	0.0004
LAD maximum aneurysm (mm)	9.3 ± 2.3	9.8 ± 3.7	0.5466
LCx maximum aneurysm (mm)	6.8 ± 3.1	8.9 ± 4.2	0.3114
LAD calcification (degree: 0/1/2/3)	5/3/1/8	1/2/0/2	0.7264
LCx calcification (degree: 0/1/2/3)	9/2/0/6	1/2/0/2	0.2671
RCA stenosis/occlusion	15/22	5/6	0.6399
MACE	2/22	1/6	0.5229

MFR, myocardial flow reserve; LAD, left anterior descending artery; KD, Kawasaki disease; PET, positron emission tomography; exam, examination; BP, blood pressure; MBF, myocardial blood flow; LCx, left circumflex artery; RCA, right coronary artery; MACE, major adverse cardiac event.

**Table 2 jcm-11-01134-t002:** Analysis by MFR of the LCx.

	LCx MFR ≥ 2.0	LCx MFR < 2.0	*p* Value
Cases	24	4	
Sex (M/F)	18/6	6/0	
Age of KD onset (months)	32.0 ± 21.6	1105 ± 9.6	0.0707
Age of PET Exam (months)	265.0 ± 100.1	194.3 ± 70.9	0.2372
Period from KD onset to PET Exam (months)	232.6 ± 105.9	182.8 ± 51.2	0.3087
BMI	20.2 ± 2.8	21.0 ± 2.3	0.5328
At Rest			
systolic BP (mmHg)	98.9 ± 11.8	95.5 ± 18.1	0.3927
diastolic BP (mmHg)	52.6 ± 8.9	51.5 ± 4.9	0.6215
LAD MBF (mL/min/g)	1.00 ± 0.37	1.06 ± 0.26	0.3580
LCx MBF (mL/min/g)	0.83 ± 0.18	0.99 ± 0.33	0.3934
LAD vascular resistance	56.4 ± 15.9	50.2 ± 8.7	0.4701
LCx vascular resistance	65.1 ± 13.5	55.8 ± 17.1	0.2644
Adenosine Stress			
systolic BP (mmHg)	95.4 ± 3.8	91.3 ± 9.2	0.6222
diastolic BP (mmHg)	46.8 ± 15.0	37.4 ± 6.8	0.1225
LAD MBF (mL/min/g)	2.66 ± 1.05	1.61 ± 0.75	0.0489
LCx MBF (mL/min/g)	2.31 ± 0.54	1.69 ± 0.57	0.0878
LAD vascular resistance	19.5 ± 8.5	25.3 ± 6.5	0.1680
LCx vascular resistance	21.2 ± 1.7	23.5 ± 4.1	0.3934
Coronary vascular resistance ratio (rest /adenosine stress)			
LAD	3.27 ± 1.39	2.03 ± 0.25	0.0105
LCx	3.41 ± 1.21	2.39 ± 0.28	0.0418
LAD MFR	2.72 ± 0.60	1.47 ± 0.43	0.0025
LAD maximum aneurysm (mm)	9.6 ± 2.6	8.5 ± 2.6	0.8057
LCx maximum aneurysm (mm)	7.2 ± 3.6	7.4 ± 2.9	0.8384
LAD calcification (degree: 0/1/2/3)	5/3/1/9	1/2/0/1	0.5156
LCx calcification (degree: 0/1/2/3)	7/2/0/7	1/2/0/1	0.1877
RCA stenosis/occlusion	17/24	3/4	1.0000
MACE	3/24	0/4	1.0000

MFR, myocardial flow reserve; LCx, left circumflex; KD, Kawasaki disease; PET, positron emission tomography; Exam, examination; BP, blood pressure; MBF, myocardial blood flow; LAD, left anterior descending artery; RCA, right coronary artery; MACE, major adverse cardiac event.

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
