# Peer review of "Peripheral Coronary Artery Circulatory Dysfunction in Remote Stage Kawasaki Disease Patients Detected by Adenosine Stress 13N-Ammonia Myocardial Perfusion Positron Emission Tomography"

_jcm, 2022, doi:10.3390/jcm11041134_

Round 1

Reviewer 1 Report

I cordially revised your manuscript according to the context, logics, and grammar.

Please check the attached file. (Red dashed parts are to be deleted, and yellow parts are to be changed.)

Author Response

Reviewer 1

Correspondence to Reviewer 1

Response: Thank you for your valuable comments and for pointing out the serious errors.

We have checked and corrected them.

Line 61

Response: Thank you for your suggestion. I changed it to "The clinical characteristics of enrolled patients were examined and compared after dividing them into two groups:"

Line 65

 I think body mass index, instead of body surface area, would be a more significant variable. So I recommend to use BMI, when comparing two groups.

Response: We changed it to “Body Mass Index (BMI)”. All other instances of BSA were also changed to BMI.

Line 68

Response: Changed to “adverse”.

Line 76

“Adenosine loading” was changed to “Adenosine stress

Line 103-105

Response: Changed to “between the stress and resting states” and “the diastolic blood pressure of the median artery measured at rest and during adenosine stress was substituted as an approximation of the coronary blood pressure.

Line 107

Please check if the formula is correct. I think "vascular resistance= diastolic pressure/MBF" would be correct.

Response: Thank you very much for pointing out a very serious error. Yes, you are correct.

Changed to “LAD vascular resistance = diastolic pressure/LAD MBF; LCx vascular resistance = diastolic pressure/LCx MBF

Line 119

Response: Changed to “The chi-square test was used

Line 128 and 130

Added the word ”median

Line 130

Added “The median BMI was 20.2 (16.2, 24.9)

Line 133 and 134

Changed to “in the LAD group” and “those in the LCX group

Line 140

Changed to “compared between

Line 143

Changed to “The analysis of MFR of the LAD

Line 146

Added “compared with the abnormal MBF group

In Table 1 & 2, "vascular resistance ratio (adenosine loading/rest)" must be corrected to "vascular resistance ratio (rest/adenosine loading)".

Response: Yes, you are correct. Thank you for pointing out this serious mistake. I corrected the error at all relevant instances.

Line 149: These data represent vascular resistance ratio, not vascular resistance. Please check them again and correct them.

Response: Yes, thank you again for your correct suggestion. I changed it to the correct numbers.

Line 159, 181, 228

I think body mass index, instead of body surface area, would be a more significant variable. So I recommend to use BMI, when comparing two groups.

Response: All instances of BSA were changed to BMI.

Line 199, 238

In Table 1 & 2, "vascular resistance ratio (adenosine loading/rest)" must be corrected to "vascular resistance ratio (rest/adenosine loading)".

Response: Thank you for your very important suggestion. I corrected all the error at all the relevant instances.

Line 274

Response: Changed to “coronary vascular resistance

Line 276-280

Changed to “No significant differences were detected in the MBF and coronary vascular resistance in the LCx under adenosine stress between the two groups. MBF in the LAD was significantly higher in the normal MFR group of LCx under adenosine stress (LAD: 2.66±1.05 vs. 1.61±0.75 ml/g/min, p=0.0489) than in the abnormal MFR group, but there was no significant difference in coronary vascular resistance.

Line 281

Changed to “significantly decreased

Line 285

Changed to “younger”.

Line 288

Changed to “BMI

Line 298

Changed to “is significantly decreased,

Line 310

This should be based on the data.

Response: We intended to say disturbance of microcirculation extended to the whole LCA because LCx coronary resistance is significantly decreased in the LAD-MFR abnormal group, and vice versa. Therefore, we changed the sentence to read, “and the disturbance extended to the total left coronary artery (LCA)”.

Line 325

Changed to “branches

Line 333

Changed to “tended

Line 334

Changed to “severe”.

Reviewer 2 Report

Peripheral Coronary Artery Circulatory Dysfunction in Remote 2 Stage Kawasaki Disease Patients Detected by Adenosine Load- 3 ing 13N-Ammonia Positron Emission Tomography

method

All patients had undergone coronary angiography or coronary computed tomography (CT) within 2 years of this examination, and those with morphological stenosis of 75% or more in the left anterior descending (LAD) or circumflex (LCx) branches were excluded from this study.  Myocardial blood flow was determined by Adenosine Load- 3 ing 13N-Ammonia Positron Emission Tomography

Right coronary artery disease was not studied only left coronary heart disease was studied to detect small vessel disease in the contralateral vessel. 

The introductory statement could read better

 A problem with coronary artery lesions in the remote stage of Kawasaki disease (KD) is the appearance of stenotic lesions due to ongoing vascular remodeling.

Suggest it be replaced by

Kawasaki disease is associated with acute and latent remodeling of proximal coronary vessels.  Distal microcirculation may undergo remodeling compromising myocardial perfusion.

Since FFR and CFR were not performed in this patient population, the discussion of these entities is not relevant. Further the affect of coronary aneurysm on these techniques is not well described. PET is an established tool for myocardial perfusion and is sufficient to prove the authors point that the microcirculation has adverse latent effects from Kawasaki disease.

Line 192 “mobility” should be replaced by “morbidity”

Mace needs to be reported what were the events.  There were not very many so they should be reported.

The paper seemed more complicated than necessary and was difficult to read.  The information is important demonstrating abnormalities in microvasculature even in contralateral vessel.  The discussion should center on pathophysiology.  Is this continue vasculitis or failure to repair damaged endothelium?  Therapies to correct the disorder would be different depending on the pathophysiology.  The FFR and CFR discussion since it was not performed confuses the findings more

Author Response

Correspondence to Reviewer 2

Peripheral Coronary Artery Circulatory Dysfunction in Remote 2 Stage Kawasaki Disease Patients Detected by Adenosine Load- 3 ing 13N-Ammonia Positron Emission Tomography

method

All patients had undergone coronary angiography or coronary computed tomography (CT) within 2 years of this examination, and those with morphological stenosis of 75% or more in the left anterior descending (LAD) or circumflex (LCx) branches were excluded from this study.  Myocardial blood flow was determined by Adenosine Load- 3 ing 13N-Ammonia Positron Emission Tomography

Right coronary artery disease was not studied only left coronary heart disease was studied to detect small vessel disease in the contralateral vessel. 

The introductory statement could read better

 A problem with coronary artery lesions in the remote stage of Kawasaki disease (KD) is the appearance of stenotic lesions due to ongoing vascular remodeling.

Suggest it be replaced by

Kawasaki disease is associated with acute and latent remodeling of proximal coronary vessels.  Distal microcirculation may undergo remodeling compromising myocardial perfusion.

Response: Thank you for your valuable comments and suggestion. However, at this time, there is not enough evidence on coronary microcirculation in the remote stage of Kawasaki disease. We have added the following sentence to the introduction based on your comments.

Line 47-48: As decreased MFR was reported in remote stage KD patients[5,6], distal microcirculation may undergo remodeling and compromise myocardial perfusion in the remote stage of KD.

Since FFR and CFR were not performed in this patient population, the discussion of these entities is not relevant. Further the affect of coronary aneurysm on these techniques is not well described. PET is an established tool for myocardial perfusion and is sufficient to prove the authors point that the microcirculation has adverse latent effects from Kawasaki disease.

Response: As you pointed out, we did not measure FFR and CFR by catheterization in this study. However, I think it is important to consider the significance of the two test methods as a way to assess coronary microcirculation. Particularly in adult patients, FFR has become clinically accepted as the criterion for PCI intervention, and I think it is necessary to discuss FFR and CFR in order to rethink the usefulness of CFR. Also, clinically, CFR and MBF have the same significance, so we are discussing them based on the reports of CFR to date. For this reason, we do not think this discussion is unnecessary.

As you pointed out, there are not many reports on FFR and CFR in Kawasaki disease coronary aneurysms, but our group has previously reported on the importance of FFR and CFR; accordingly I added the following sentence to the Introduction.

Line 40-41; We have previously reported how FFR and CFR were useful for detecting myocardial ischemia in children with KD cardiac sequelae [3]

Line 298 “mobility” should be replaced by “morbidity”

Response: Thank you for your suggestion. We have corrected it.

MACE needs to be reported what were the events.  There were not very many so they should be reported.

Response: Thank you for pointing this out. All three cases were acute myocardial infarction, and I have inserted this into Line 131.

The paper seemed more complicated than necessary and was difficult to read.  The information is important demonstrating abnormalities in microvasculature even in contralateral vessel.  The discussion should center on pathophysiology.  Is this continue vasculitis or failure to repair damaged endothelium?  Therapies to correct the disorder would be different depending on the pathophysiology.  The FFR and CFR discussion since it was not performed confuses the findings more

Response: Although it was stated in the Study Limitation that RCA was not considered, the Study limitation has been revised as follows.

Only LCAs could be assessed in this study. There were only three cases with RCA MFR<2.0. In addition, quantitative analysis of RCA territory often demonstrated apparently erroneous values due to overflow from the tracer in the blood pool of the right ventricle or due to artifacts from respiratory motion. For this reason, we could not include RCA in the analysis.

As shown in this study, it is presumed that damage to the peripheral coronary arteries may extend to the entire myocardium; hence, it will be necessary to accumulate more cases and to improve the accuracy of analysis and examine abnormalities of the MFR in the RCA territory separately in the future.

The pathology reported in the remote stage of Kawasaki disease is post-inflammatory scarring. There is little infiltration of inflammatory cells such as macrophages, and the acute stage inflammation is not persistent. However, it is the transformed vascular smooth muscle cells, fibroblasts, and the extracellular matrix they produce that are central to the intimal thickening that leads to regression of the aneurysm. In particular, transformed vascular smooth muscle cells continue to produce a variety of growth factors in the remote phase, leading to continued active vascular remodeling. In these coronary arteries, impairment of vascular endothelial cell function continues to be observed, and it is suspected that the presence of impaired vascular endothelial cell function is a risk factor for the development of atherosclerosis in younger patients. There is still no evidence for treatment in the remote phase, except for thromboprophylaxis of the aneurysm causing AMI. Recent guidelines for treatment of the remote phase of KD include recommendations for statins and ACE inhibitors due to concerns about early atherosclerosis risk, but there is as yet no level of evidence.

This is what has been reported in remote stage coronary aneurysms in Kawasaki disease. Since this paper demonstrates the existence of peripheral circulatory disturbance, which has not been reported before, we discuss FFR and CFR, which are methods for evaluating peripheral vascular function. We do not consider this to confuse the discussion.

The following sentence has been added to Discussion.

Line 311-314; Although there are many reports of continued endothelial cell dysfunction in the remote stage of Kawasaki disease[13], there are few reports that directly demonstrate peripheral circulatory dysfunction. Therefore, we believe this paper is particularly valuable.

Round 2

Reviewer 2 Report

I appreciate your changes and the paper can be published as written. My opinion the cfr ffr discussi8on is an attempt to make the paper relevant but without performing these tests it adds nothing to the paper.  I think what you are trying to say is that 40 years of concern over blockage is being replaced by more predictive tools of coronary perfusion. ffr cfr and your pet study is a surogot for these tests.

Author Response

Thank you very much for the meaningful peer review, which has greatly improved our paper.

As you pointed out, I have added the following sentences to the Study Limitation for CFR and FFR. (Lines 348-353)

The Doppler wire (FloWire XT, Cardiometrics Inc.), required for CFR evaluation, is currently difficult to obtain, and without doppler wire, CFR cannot be performed in many cases. Thus, it is believed that CFR will be replaced by MFR in the future. Furthermore, FFR, which evaluates stenosis in the epicardial artery, was not examined; however, since patients with stenosis of 75% or more on coronary angiography were excluded, it is unlikely to affect the results of this study.